# The Effect of Vitamin D Supplementation on the Severity of Symptoms and the Quality of Life in Irritable Bowel Syndrome Patients: A Systematic Review and Meta-Analysis of Randomized Controlled Trials

**DOI:** 10.3390/nu14132618

**Published:** 2022-06-24

**Authors:** Mohamed Abuelazm, Shoaib Muhammad, Mohamed Gamal, Fatma Labieb, Mostafa Atef Amin, Basel Abdelazeem, James Robert Brašić

**Affiliations:** 1Faculty of Medicine, Tanta University, Tanta 31527, Egypt; dr.mabuelazm@gmail.com (M.A.); moh.gamal762@gmail.com (M.G.); 2Department of Internal Medicine, Gulab Devi Hospital, Lahore 54000, Pakistan; shoaibposwal@live.com; 3Faculty of Medicine, Beni-Suef University, Beni-Suef 62511, Egypt; fatmalabieb30@gmail.com; 4Faculty of Medicine, Cairo University, Cairo 12613, Egypt; atef.mostafa20160622@gmail.com; 5Department of Internal Medicine, McLaren Health Care, Flint, MI 48532, USA; baselelramly@gmail.com; 6Department of Internal Medicine, Michigan State University, East Lansing, MI 48823, USA; 7Section of High Resolution Brain Positron Emission Tomography Imaging, Division of Nuclear Medicine and Molecular Imaging, The Russell H. Morgan Department of Radiology and Radiological Science, The Johns Hopkins University School of Medicine, Baltimore, MD 21287, USA

**Keywords:** alternative intervention, calciferol, confidence interval, flow chart, gastrointestinal disorder, heterogeneity, mean difference, placebo, protocol, random

## Abstract

Irritable bowel syndrome (IBS), a gastrointestinal disorder affecting 7–12% of the population, is characterized by abdominal pain, bloating, and alternating bowel patterns. Data on risk and protective influences have yielded conflicting evidence on the effects of alternative interventions, such as vitamin D. This review focuses on the effects of vitamin D on IBS. A systematic review and meta-analysis considered all articles published until 4 April 2022. The search for randomized controlled trials assessing vitamin D efficacy in IBS with outcomes, primary (Irritable Bowel Severity Scoring System (IBS-SSS)) and secondary (IBS quality of life (IBS-QoL) and serum level of calcifediol (25(OH)D)), was performed on six databases, Google Scholar, Web of Science, SCOPUS, EMBASE, PubMed (MEDLINE), and Cochrane Central Register of Controlled Trials. We included six trials with 616 patients. The pooled analysis found no difference between vitamin D and placebo in improving IBS-SSS (MD: −45.82 with 95% CI [−93.62, 1.98], *p* = 0.06). However, the pooled analysis favored vitamin D over placebo in improving the IBS-Qol (MD: 6.19 with 95% CI [0.35, 12.03], *p* = 0.04) and serum 25(OH)D (MD: 25.2 with 95% CI [18.41, 31.98], *p* = 0.00001). Therefore, further clinical trials are required to reach clinically applicable and generalizable findings.

## 1. Introduction

Irritable bowel syndrome (IBS), a common gastrointestinal condition affecting 7–12% of the general population [1], is characterized by fluctuating severity of symptoms, including abdominal discomfort, pain, bloating, and alternating bowel habits. Although the pathogenesis is unclear, an association with psychological stress has long been identified, ultimately causing disturbed social functions and poor quality of life [2]. Despite the significant burden of the disease, no treatment has proven to be entirely effective [3].

Given the lack of established therapies, a large proportion of the patients do not show improvement on conventional treatments [4]. Therefore, around one-third of the patients opt for alternative therapies [5]. Since vitamin D deficiency is a common condition [6,7,8,9] associated with cardiovascular disease (CVD) [10], cancer [11], insulin resistance [12,13,14], and other chronic diseases [15], investigators have sought to identify possible associations of vitamin D deficiency with IBS. Therefore, studies have shown an improvement of IBS symptoms with the administration of vitamin D [16,17], an agent playing a critical role in calcium and phosphorous metabolism and homeostasis as well as anti-inflammatory and immunomodulatory activities [18]. 

Worldwide, a 30–50% prevalence for vitamin D deficiency has been estimated across different demographic groups [19]. Since vitamin D is essential in maintaining mucosal surfaces like the intestinal mucosal barrier [20], vitamin D deficiency presents with gastrointestinal symptoms led by mucosal damage [20,21]. A comparative case-control study demonstrated a significant relationship between vitamin D deficiency and IBS with vitamin D deficiency in 82% of the IBS patients compared to 32% of healthy participants [22]. Additionally, several studies report vitamin D deficiency in people with IBS [23,24]. Furthermore, various psychiatric disorders such as depression and anxiety are associated with vitamin D deficiency, which in turn shows an association with IBS [25,26]. 

Accordingly, the small number of randomized controlled trials (RCTs) to evaluate the effect of vitamin D in IBS patients yield contradictory findings [27,28,29,30,31,32]. Vitamin D supplementation randomized controlled trials (RCTs) demonstrated improved IBS severity of symptoms (IBS-SSS) in some RCTs [27,28,30,32], but not in others [29,31], and improved IBS quality of life (IBS-QoL) in some RCTs (27–29), but not in another [31]. Therefore, in order to resolve these controversies, we conducted a systematic review and meta-analysis to evaluate the effect of vitamin D supplementations on the severity of symptoms and the quality of life in IBS patients.

## 2. Materials and Methods

### 2.1. Protocol Registration

We registered and published our review protocol in PROSPERO with ID: CRD42022323299. We strictly performed this systematic review and meta-analysis according to the Preferred Reporting Items for Systematic Reviews and Meta-Analyses (PRISMA) statement [33] and the Cochrane Handbook of Systematic reviews and meta-analysis [34].

### 2.2. Data Sources & Search Strategy

Two reviewers (B.A. and M.A.) systematically searched the following electronic databases: Google Scholar, Web of Science (WOS), SCOPUS, EMBASE, PubMed (MEDLINE), and Cochrane Central Register of Controlled Trials until 4 April 2022. We did not use any search filters. The detailed search strategy and results are demonstrated in (Table 1). 

### 2.3. Eligibility Criteria

We included RCTs with the following PICO criteria: Population (P): irritable bowel syndrome (IBS) patients regardless of disease type, grade, and serum vitamin D baseline level; Intervention (I): vitamin D supplementations regardless of dose and treatment duration; Control (C): placebo; Outcomes (O): primary outcome: IBS severity scoring system (IBS-SSS) [35]. Our secondary outcomes are quality of life assessed by a self-reported specific IBS-related quality of life questionnaire (IBS-Qol) [36] and serum level of calcifediol (25(OH)D) (https://pubchem.ncbi.nlm.nih.gov/substance/53789608) (accessed on 1 June 2022).

The IBS-SSS is a questionnaire validated for application for IBS patients to determine the burden of the disease; during 10 days, IBS-SSS assesses the following (abdominal pain severity, abdominal pain frequency, abdominal distention or tightness severity, bowel habits dissatisfaction, interference with life in general). Each item is scored on a scale from 0 to 100 and with a range of (0–500). Mild, moderate, and severe cases are indicated by scores of 75–175, 175–300, and >300, respectively, and a score reduction of 50 or more is considered clinically significant [35]. The (IBS-Qol) questionnaire included dysphoria, interference with activity, body image, health worry, food avoidance, social reaction, sexual, and other issues among eight sub-scales and 34 items with a score range from 0 to 100; the higher the score the better Qol [36].

The exclusion criteria involved pilot studies, animal studies, in vitro studies (tissue and culture studies), observational studies (case-control, cross-sectional, case series, and report), press articles, editorial letters, conference abstracts, registered protocols, and book chapters. 

### 2.4. Study Selection

Using Covidence online software [37], two reviewers (M.G. and F.L.) independently assessed the titles and abstracts of the retrieved records and then the full-text articles for the previous eligibility criteria. A third reviewer (M.A.) resolved any conflicts. The selection process is demonstrated in a PRISMA flow chart [33] (Figure 1). 

### 2.5. Data Extraction

Two reviewers (M.G. and F.L.), using a pre-designed and pilot tested extraction sheet, independently extracted the following: study characteristics (first author name, year of publication, country, and study design); baseline information (age, gender, vitamin D dose, treatment duration, diagnostic, and IBS sub-type, disease duration, and severity); outcomes data (IBS-SSS [35], IBS-Qol [36], and serum 25(OH)D3). A third reviewer (M.A.) resolved any conflicts.

We sought the extraction of mean difference (MD) and standard deviation (SD) for continuous variables. If the data was not available, we used the relevant formulas [34,38] to calculate the MD and SD either from the median, 25th, and 75th percentiles [30] or calculated SD from standard error [29].

### 2.6. Risk of Bias and Quality Assessment

Two reviewers (M.A.A. and F.L.) independently assessed the included studies for the risk of bias (ROB) using The Cochrane Collaboration’s tool for assessing the risk of bias in randomized trials [39]), based on the following domains: random sequence generation (selection bias), allocation concealment (selection bias), blinding of participants and personnel (performance bias), blinding of outcome assessment (detection bias), incomplete outcome data (attrition bias), selective reporting (reporting bias), and other potential sources of bias. Conflicts were resolved by discussion. Two reviewers (M.A.A. and F.L.) used the Grading of Recommendations Assessment, Development and Evaluation (GRADE) Working Group recommendation [40,41,42] for quality of evidence assessment. We considered inconsistency, imprecision, indirectness, publication bias, and risk of bias. Our conclusions on the quality of evidence were justified, recorded, and included in the results reporting for each outcome. A third reviewer (M.A.) resolved any conflicts.

### 2.7. Statistical Analysis

The statistical analysis was conducted using RevMan v5.3 software [43]. The effects of vitamin D supplementations on the continuous outcome variables were estimated by comparing the pooled MD and SD of changes before and after the treatment in the treatment group with those in the control group. The pooled MD was synthesized using the Mantel–Haenszel method. All data were presented with the *p*-value and corresponding 95% confidence interval (CI), and a *p*-value < 0.05 was considered statistically significant. We assessed heterogeneity by visual inspection of the forest plots and evaluated by I-square and chi-square tests. The chi-square test determines whether there is significant heterogeneity. In contrast, the I-square evaluates the magnitude of heterogeneity. According to the Cochrane Handbook (chapter nine) [34], an alpha level below 0.1 is considered to be a significant heterogeneity (for the chi-square test), and the I-square test is interpreted as follows: (0–40%: might not be important; 30–60%: may represent moderate heterogeneity; 50–90%: may represent substantial heterogeneity). We used the random effects model in case of significant heterogeneity and the fixed effects model otherwise. 

We excluded one study at a time and repeated the analysis to perform the sensitivity analysis to assess the impact of each study on the overall study effects size of the outcome. In addition, we performed a sub-group analysis based on the vitamin D dose and diagnostic criteria to test the stability of our results. We did not perform funnel plots to indicate the publication bias because we included less than ten studies [44]. 

## 3. Results

### 3.1. Search Results and Study Selection

After searching databases, we retrieved 1627 records. We excluded 282 duplicates, leaving 1345 for the title and abstract screening. Then we excluded 1320 records, leaving 25 full-text articles to be screened. Finally, we included six articles [27,28,29,30,31,32] in our systematic review and meta-analysis. The screening process is illustrated in a PRISMA flow chart (Figure 1). 

### 3.2. Characteristics of Included Studies

We included six RCTs [27,28,29,30,31,32] with a total population of 616 participants including 310 in the vitamin D group and 306 in the placebo group. The mean age was 34.6 ± 9.12 years for the vitamin D group and 33.2 ± 8.3 years for the placebo group. The females represented 137/212 (64.6%) in the vitamin D group and 135/208 (64.9%) in the placebo group. The duration of treatment ranged from 1.5 up to 6 months. Three studies were conducted in Iran [27,29,30], two in Egypt [28,32], and one in the UK [31] (Table 2). Three studies used ROME III criteria for IBS diagnosis [27,28,29], and another three used ROME IV criteria [30,31,32]. 

### 3.3. Risk of Bias and Quality of Evidence

All the included trials were low risk of selection, performance, and detection biases. Five trials [28,29,30,31,32] exhibited a low risk of attrition bias, but one [27] showed a high risk of attrition bias due to unbalanced and unexplained loss of follow-up between the two arms. Four trials [27,29,30,31] exhibited a low risk of reporting and other biases, but two trials [28,32] were unclear due to the lack of published protocol (Figure 2). The quality of evidence using the GRADE system [40,41,42] is demonstrated in Table 3. 

### 3.4. Primary Outcome

#### IBS-SSS 

The pooled mean difference showed no difference between vitamin D and placebo (MD: −45.82 with 95% CI [−93.62, 1.98], *p* = 0.06) (very low-quality evidence). Pooled studies were not homogenous (*p* = 0.00001, I-square = 95%) (Figure 3A, Table 3). To resolve heterogeneity, we conducted a sensitivity analysis, excluding one study in each scenario. However, heterogeneity was not resolved by sensitivity analysis (Table 4). Sensitivity analysis showed similar effect except after removing the study of Jalili et al., 2019 [29] and Williams et al., 2021 [31]; the pooled mean difference favored vitamin D over placebo ((MD: −59.82 with 95% CI [−111.85, −7.79], *p* = 0.02)–(MD: -57.82 with 95% CI [−110.94, −4.71], *p* = 0.03)), respectively (Table 4).

We conducted ag-group analysis based on the following: A- Vitamin D3 dosage, to evaluate the effect of 50.000 IU vitamin D3 dose and dosage less than 50.000 IU vitamin D3 on IBS-SSS; pooled mean difference showed no difference between vitamin D and placebo ((MD:−24.2 with 95% CI [−65.79, 17.39], *p* = 0.25)–(MD: −67.07 with 95% CI [−164.66, 30.53], *p* = 0.18)) respectively (Figure 4A). B- IBS diagnostic criteria, pooled mean difference showed no difference between vitamin D and placebo either in IBS patients diagnosed with ROME III criteria (MD: −26.18 with 95% CI [−65.96, 13.61], *p* = 0.2) or ROME IV criteria (MD: −66.08 with 95% CI [−170.59, 38.42], *p* = 0.22) (Figure 4B).

### 3.5. Secondary Outcomes

#### 3.5.1. IBS-Qol

The pooled mean difference favored vitamin D over placebo (MD: 6.19 with 95% CI [0.35, 12.03], *p* = 0.04) (very low-quality evidence). Pooled studies were not homogenous (*p* = 0.03, I-square = 66%). (Figure 3B, Table 3) To resolve heterogeneity, we conducted a sensitivity analysis, excluding one study in each scenario. Heterogeneity was best resolved by excluding the study of El Amrousy et al. [27] (*p* = 0.98, I-square = 0%). The pooled mean difference favored vitamin D over placebo (MD: 3.26 with 95% CI [2.14, 4.39], *p* = 0.00001) (Figure 5, Table 4).

#### 3.5.2. Serum 25(OH)D

The pooled mean difference favored vitamin D over placebo (MD: 25.2 with 95% CI [18.41, 31.98], *p* = 0.00001) (moderate-quality evidence). Pooled studies were not homogenous (*p* = 0.00001, I-square = 93%). (Figure 3C, Table 3) To resolve heterogeneity, we conducted a sensitivity analysis by excluding one study in each scenario. However, heterogeneity was not resolved by sensitivity analysis (Table 4).

## 4. Discussion

In our meta-analysis, we evaluated the efficacy of vitamin D supplementation to determine whether it causes any improvements in symptoms of IBS. Our analysis showed that vitamin D supplementation failed to improve IBS symptoms. However, the IBS-QoL score was improved after the treatment with vitamin D.

Our analysis found no difference between vitamin D and placebo in improving IBS-SSS. However, four RCTs [27,28,30,32], two in Iran and two in Egypt, showed an improvement in IBS symptoms, while two RCTs [29,31] showed no preference for vitamin D over the placebo. Our dosage sub-group analysis found no difference between high dosages (50,000 IU) and lower dosages (less than 50,000 IU). This surprising finding indicates no dose-response relationship between vitamin D and reducing IBS-SSS. However, the treatment duration of the RCTs that used both dosages varies, which may significantly impact our findings. Regarding IBS-Qol, our pooled analysis favored vitamin D over placebo. Our results are consistent with three RCTs [27,28,29] while only one study by Williams et al. [31] showed no improvement in IBS-QoL.

Moreover, pooled analysis favored vitamin D over placebo regarding increasing the serum level of 25(OH)D. In all the included studies, the serum level of vitamin D was higher in the vitamin D group compared to the placebo. Williams et al. [31] showed no difference in QoL between the vitamin D group and placebo. The contradictory results of Williams et al. [31] regarding IBS-SSS and IBS-Qol can be attributed to a few critical differences in the study design that set it apart from the rest of the included RCTs. First, Williams et al. [31] used community-based sampling rather than outpatient and institution-based selection in other RCTs. Second, Williams et al. [31] had the largest population size compared to the rest of the studies. Finally, the regional difference might have also played a role as all the vitamin D supporting studies were based in the Middle East, compared to the UK-based RCT by Williams et al. [31]. Moreover, Jalili et al. [29] failed to show any preference for vitamin D administration in improving IBS-SSS. This can be explained by the short duration of vitamin D treatment [29]. Therefore, after excluding Williams et al. [31] or Jalili et al. [29], our pooled analysis advocated using vitamin D over the placebo.

The reason behind the effect of vitamin D on symptoms and quality of life in IBS patients is still to be investigated. To clarify, IBS is known to have complex pathophysiology, and both peripheral and central factors have been suggested to play critical roles [45]. Vitamin D can impact the gastrointestinal system’s health due to its immunomodulatory characteristics [3]. In the pathogenesis of IBS, immunological and inflammatory roles are of particular importance as literature has emphasized the activation of the inflammatory mediator to have a crucial part in the IBS development [46]. The upregulated mast cells, T-cells, and other pro-inflammatory cytokines are among the few known key players [2,45,47]. The increased inflammation upregulates the neural activity in the intestine leading to visceral hypersensitivity and worsened feeling of abdominal pain [47]. Given the anti-inflammatory effects of vitamin D, its usage can improve intestinal inflammation [48]. Therefore, decreased inflammation can improve the sensory nervous system in the gut, causing normalization of gut functionality and managing IBS symptoms [3]. This can be explained by the presence of vitamin D receptors in the nervous system, where they play a role in the synthesis, maintenance, and upregulation of neurotransmitters levels [49,50]. Accordingly, the administration of vitamin D can improve IBS symptoms, which is contradictory to our findings.

Another leading etiology of IBS is psychological factors. Anxiety and depression have long been implicated in the development of IBS, and studies have shown a direct connection between depression and the associated bowel-related symptoms [2]. Many patients with IBS reported the presence of some psychological symptoms before their IBS started [4]. Furthermore, psychological stress can cause alteration in bowel patterns, and the treatment of stress and anxiety show a remarkable improvement in IBS symptoms [2]. In addition, literature shows that patients with depression are typically vitamin D deficient; therefore, vitamin D administration can improve their depressive symptoms [49,51]. Hence, vitamin D supplementation also applies in managing this etiology and improving the quality of life in accordance with our findings.

Regarding the safety and tolerability of vitamin D in IBS, vitamin D is absorbed through the enterocyte’s apical membrane, and 25(OH)D levels increase slowly and reach the peak within 7 to 14 days [52]. Vitamin D supplementation in healthy individuals should be administrated with caution. The safe therapeutic limit of vitamin D intake is 4000 IU daily to prevent side effects [53]. Exogenous vitamin D can elevate the risk of hypercalcemia and hypercalciuria. It may also cause mineral deposition in soft tissues. Severe side effects may arise when long-term dosage exceeds ten-fold the endorsed quantity. Toxicity causing vomiting, lethargy, confusion, and arrhythmia have been reported in the literature [54]. Despite using very high intermittent doses of vitamin D (50,000 IU) in three of the included RCTs [27,29,30], no adverse effects were noted. The most likely reason behind this seems to be the short duration of treatment. The three RCTs [28,31,32] (two Egyptian and one UK-based) kept the vitamin D dosage within the suggested limits. Patients show excellent tolerance when the recommended dosage is used [52]. In addition, vitamin D supplements are available over the counter with an easily affordable price tag.

In a recent systematic review and meta-analysis, Chong et al. [55] reported that vitamin D significantly improved IBS-SSS but found no difference between vitamin D and placebo in improving IBS-Qol, which is contradictory to our findings. On the one hand, the difference in IBS-SSS outcome can be attributed to the fact that Chong et al. [55] included Tazzyman et al. [56] which is a pilot study that used IBS visual analogue score (VAS-IBS) and considered underpowered to provide significant findings, hence excluded from our analysis, and Jalili et al. [57] who compared soy isoflavones and/or vitamin D. However, vitamin D was administrated along with placebo to substitute soy isoflavones which can significantly impact the findings. To clarify, patients that receive a placebo have significantly improved findings compared to baseline because IBS patients’ mental status is significantly affected by the placebo given their expectations and desire to receive therapy are more crucial than the drug’s composition [30,58]. Hence, this was excluded from our analysis. On the other hand, the difference in IBS-Qol outcome can be attributed to an error in the analysis by Chong et al. [55]. To clarify, they miss-extracted the data of Williams et al. [31] by changing the values of IBS-Qol outcome numbers from negative to positive changing the direction of the effect of the Williams et al. [31] study and their pooled analysis. We clarified the methodological flaws of Chong et al. [55] in our recently published editorial [59]. In another recent review by Haung et al. [60], vitamin D was effective in improving IBS-SSS which is also contradictory to our findings. This difference is attributed to the narrow and defective searching technique excluding three includable studies [29,31,32] especially Williams et al. [31] and Jalili et al., 2019 [29] which showed no effect of vitamin D on IBS-SSS. To conclude, our review introduces more robust findings on the effect of vitamin D in IBS patients.

### 4.1. Strengths

The major strength of our study is that we strictly followed the PRISMA statement [33] and the Cochrane Handbook of Systematic reviews and meta-analysis [34]. Moreover, we assessed the quality of evidence following GRADE guidelines [40,41,42] and prospectively registered and published our review protocol. Furthermore, we conducted a thorough analysis including sub-group analysis and sensitivity analysis to test the stability of our results.

### 4.2. Limitations

Our review has a few limitations; First, the lack of generalization of the participant populations: four RCTs conducted in the Middle East showed similar results [27,28,30,32], favoring vitamin D over the placebo. However, the UK-based RCT [31] failed to show any statistically significant association between vitamin D intake and improvement in IBS symptoms. Second, our analysis included few RCTs with a small sample size, further limiting the generalizability of our findings. Because we had fewer than ten studies, we did not perform a funnel plot [44]. Third, included RCTs used varied dosages of vitamin D with varied duration of treatment. Moreover, three RCTs [27,29,30] used an intermittent bolus dose of 50,000 IU, which is not a clinically practical approach to managing IBS [31]. Fourth, none of the included RCTs assessed the effect of vitamin D on the various IBS sub-types and severity grades with only Sikaroudi et al. [30] assessing vitamin D for the IBS-D sub-type only. Fifth, multiple confounding variables were not controlled, such as vitamin D deficiency, diet, psychiatric factors, and physical activity; each of them can significantly impact our findings. Finally, we detected a high level of heterogeneity between the included RCTs. Moreover, IBS-SSS and IBS-Qol outcomes yielded a very-low quality of evidence when tested by the GRADE quality system. Hence, the generalizability of our results is limited.

### 4.3. Implications for Future Research

Future large RCTs are required to assess the following: First, the long-term sustained effect of vitamin D supplementations in IBS patients beyond 24 weeks. Second, future trials should assess the effect of vitamin D in different grades and severity of IBS because each sub-type is a complicated disorder with a different clinical presentation and IBS treatment is generally dependent on its sub-type. Third, future trials should assess vitamin D levels in the baseline and report their results for the vitamin D deficient patients separately because vitamin D deficiency is a major confounding variable, given the significant difference in IBS-Qol between the vitamin D deficient and replete patients, and the improvement in IBS-Qol can be attributed to IBS symptoms severity improvement [27]. Fourth, we recommend the usage of a uniform dosage of vitamin D rather than impractical higher dosages to reach optimal and clinically practical vitamin D doses. Finally, baseline assessment of psychiatric, dietary, and physical factors is recommended due to its great impact on IBS pathogenesis.

## 5. Conclusions

Our systematic review and meta-analysis found uncertain evidence on the efficacy of vitamin D in improving the severity of symptoms and the quality of life in irritable bowel syndrome patients. However, we found significant statistical improvement in the IBS-Qol. Therefore, treatment with 25(OH)D may be beneficial for people with IBS [61]. Hence, the clinical usability of this meta-analysis is limited, and high-quality RCTs with larger populations for optimal reviews [62] are required before the clinical application of vitamin D in IBS can be endorsed.

## Figures and Tables

**Figure 1 nutrients-14-02618-f001:**
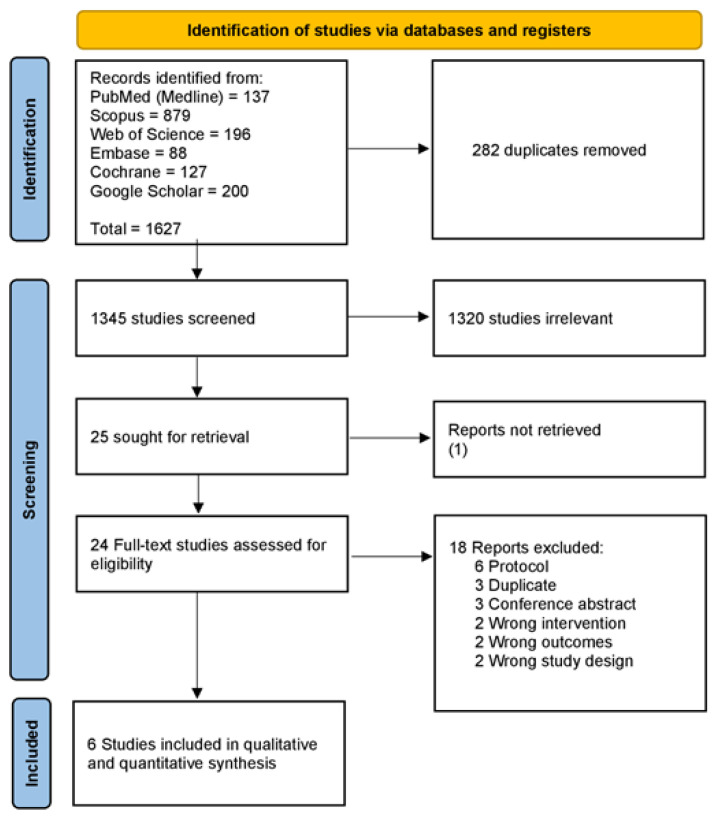
PRISMA flow chart of the screening process [33].

**Figure 2 nutrients-14-02618-f002:**
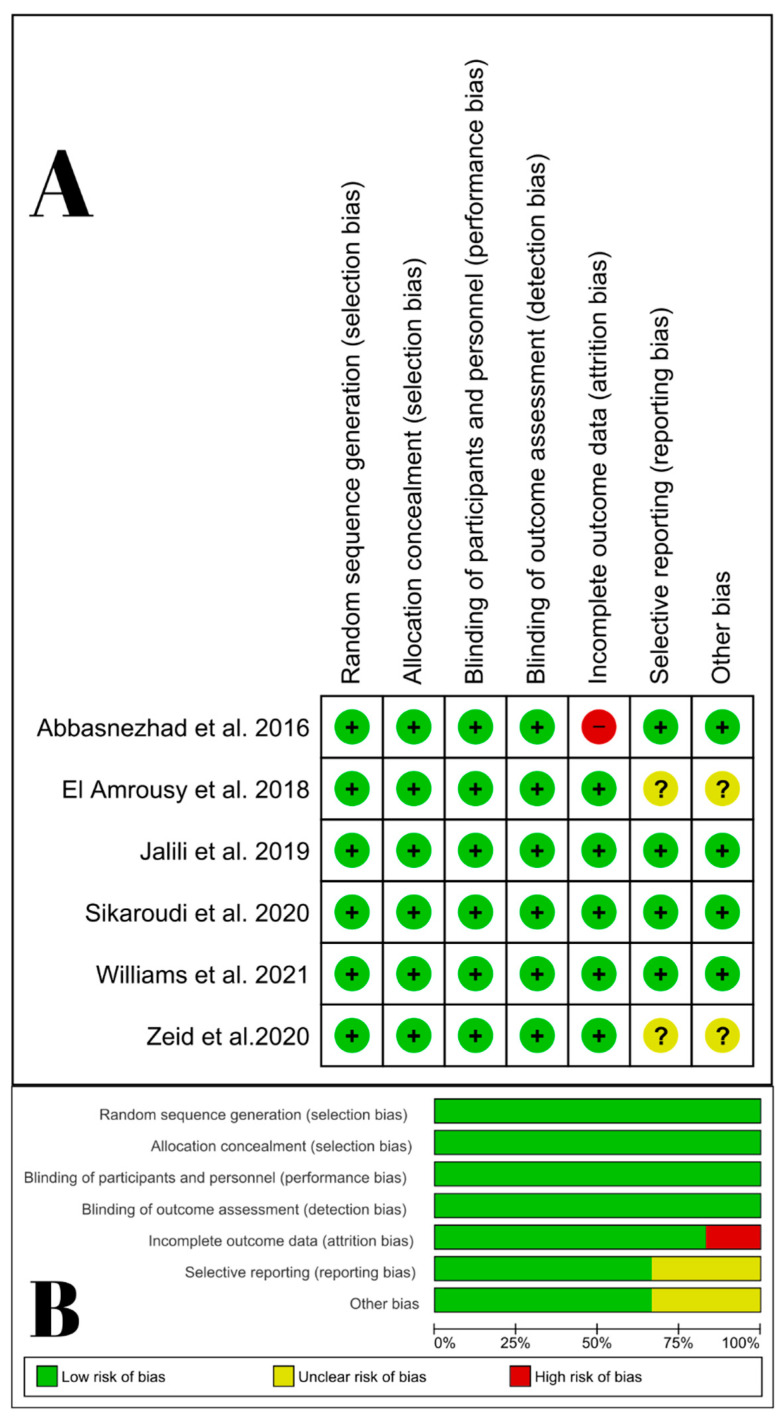
Quality assessment of risk of bias in the studies in the meta-analysis. (**A**) The upper panel presents a schematic representation of risks (low = red, unclear = yellow, and high = red) for specific types of biases of each of the studies in the review [27,28,29,30,31,32]. (**B**) The lower panel presents risks (low = red, unclear = yellow, and high = red) for the sub-types of biases of the combination of studies included in this review [43].

**Figure 3 nutrients-14-02618-f003:**
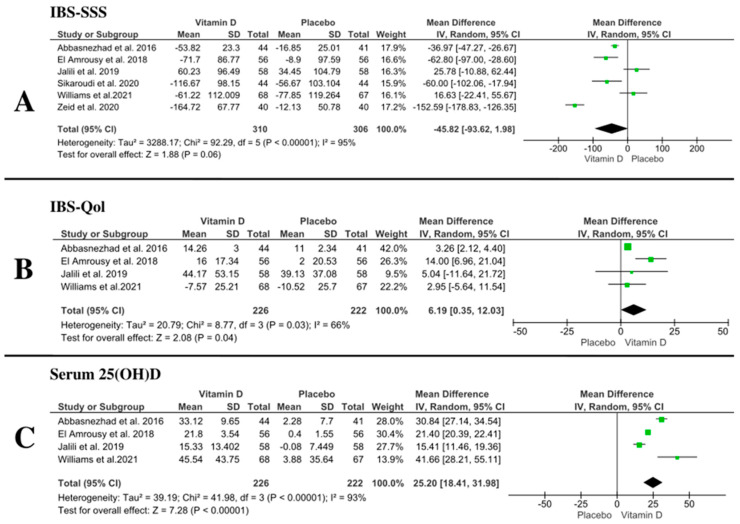
Forest plot of the included outcomes ((**A**)—IBS-SSS, (**B**)—IBS-Qol, (**C**)—serum 25(OH)D). I2: I-squared; CI: confidence interval [27,28,29,30,31,32,43].

**Figure 4 nutrients-14-02618-f004:**
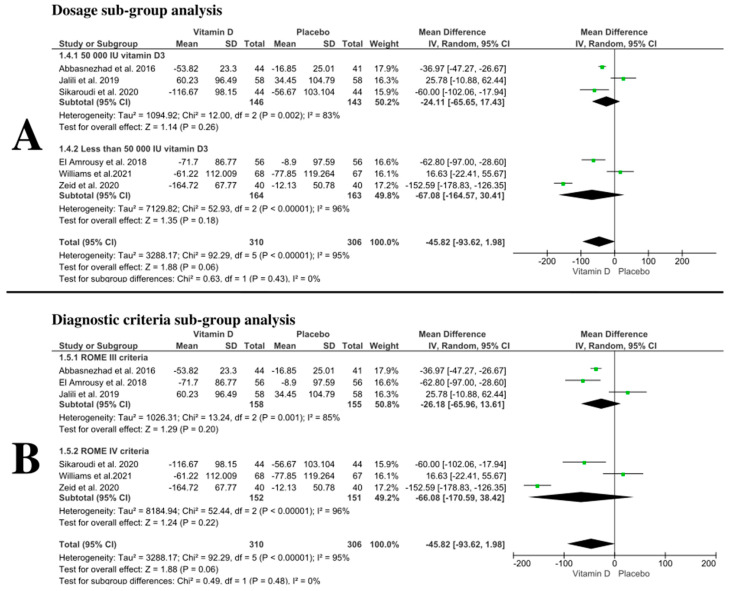
Forest plot of the sub-group analysis ((**A**)—dosage sub-group analysis, (**B**)—diagnostic criteria sub-group analysis) I2: I-squared; CI: confidence interval [27,28,29,30,31,32,43].

**Figure 5 nutrients-14-02618-f005:**
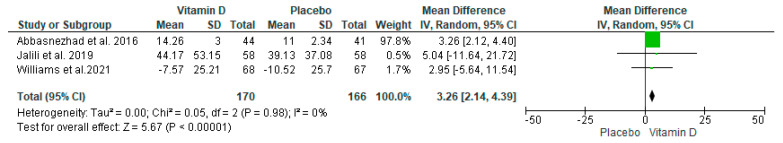
Forest plot of IBS-Qol after exclusion El Amrousy et al. [27]. I2: I-squared; CI: confidence interval.

**Table 1 nutrients-14-02618-t001:** Search terms and results in different databases.

Database	Search Terms	Search Field	Search Results
PubMed	(“Vitamin D” OR “Cholecalciferol” OR “Hydroxycholecalciferols” OR “Ergocalciferols” OR “25 Hydroxyvitamin D” OR “Dihydrotachysterol” OR “25(OH)D” OR “25-hydroxyvitamin D” OR calcifediol OR calciferol OR “Vitamin D”) AND (“Colonic Diseases, Functional”[Mesh] OR IBS OR irritable bowel syndrome OR”functional abdominal pain” OR “functional gastrointestinal” OR FGID OR “irritable colon” OR Colitis, Mucous OR Colitides, Mucous OR Mucous Colitides OR Mucous Colitis)	All Field	137
Cochrane	((Vitamin D) OR (Cholecalciferol) OR (Hydroxycholecalciferols) OR (Ergocalciferols) OR (25 Hydroxyvitamin D) (Word variations have been searched)) AND ((irritable bowel syndrome) OR (IBS) OR (functional abdominal pain) OR (Functional Colonic Diseases) OR (irritable colon) (Word variations have been searched))	All Field	127
WOS	(“Vitamin D” OR “Cholecalciferol” OR “Hydroxycholecalciferols” OR “Ergocalciferols” OR “25 Hydroxyvitamin D” OR “Dihydrotachysterol” OR “25(OH)D” OR “25-hydroxyvitamin D” OR calcifediol OR calciferol OR “Vitamin D”) AND (“Colonic Diseases, Functional”[Mesh] OR IBS OR irritable bowel syndrome OR “functional abdominal pain” OR “functional gastrointestinal” OR FGID OR “irritable colon” OR Colitis, Mucous OR Colitides, Mucous OR Mucous Colitides OR Mucous Colitis)	All Field	196
SCOPUS	(TITLE-ABSKEY ((vitamin AND d) OR (cholecalciferol) OR (hydroxycholecalciferols) OR (ergocalciferols) OR (25 hydroxyvitamin AND d) OR (dihydrotachysterol) OR (25(OH)D) OR (25-hydroxyvitamin AND d) OR (calcifediol) OR (calciferol) OR (vitamin AND d)) AND TITLE-ABS-KEY ((functional AND colonic AND diseases) OR (irritable AND bowel AND syndrome) OR (ibs) OR (functional AND abdominal AND pain) OR (functional AND gastrointestinal) OR (fgid) OR (mucous AND colitides) OR (mucous AND colitis)) AND (LIMIT-TO (DOCTYPE, “ar”))	Title, Abstract, Keywords	879
EMBASE	(“vitamin deficiency”/exp OR “vitamin deficiency” OR cholecalciferol OR hydroxycholecalciferols OR ergocalciferols OR (25 AND hydroxyvitamin AND d) OR dihydrotachysterol OR (25 AND oh AND d) OR (“25 hydroxyvitamin” AND d) OR calcifediol OR calciferol OR (vitamin AND d)) AND (“irritable colon”/exp OR “irritable colon” OR ibs OR (functional AND abdominal AND pain) OR (functional AND gastrointestinal) OR fgid OR (irritable AND colon) OR (mucous AND colitis) OR (mucous AND colitides)) AND “randomized controlled trial”/de	All Field	88
Google Scholar	(“Vitamin D” OR Cholecalciferol OR Hydroxycholecalciferol OR Ergocalciferol OR 25 “Hydroxyvitamin D”) AND (“irritable bowel syndrome” OR IBS OR “functional abdominal pain” OR “Functional Colonic Diseases” OR “irritable colon”)	All Field	1430 Ext. (first 200 only)

**Table 2 nutrients-14-02618-t002:** Characteristics of the included studies.

Study ID	Country	Study Design	Total Participants	IBS Sub-type	Follow-up Duration (Months)	Vitamin D					Placebo			
						Number	Female *n* (%)	Age (Years) Mean (SD)	Baseline Serum Vitamin D Mean (SD)	Dose	Number	Female *n* (%)	Age (Years) Mean (SD)	Baseline Serum Vitamin D Mean (SD)
Abbasnezhad et al., 2016 [27]	Iran	Single-center double blinded RCT	85	BS-D IBS-A IBS-C	6	44	28 (63.6)	37.45 (8.11)	19.65 (10.35)	50,000 IU fortnightly	41	29 (70.7)	38.45 (9.85)	18.62 (11.23)
Zeid et al., 2020 [32]	Egypt	Single-center double blinded RCT	80	N/A	3	40	N/A	37.64 (11.13)	N/A	4000 IU daily	40	N/A	38.03 (6.37)	N/A
Williams et al., 2021 [31]	UK	Single-center double blinded RCT	135	N/A	3	68	55 (80.9)	28.94 (10.03)	48.75 (27.91)	3000 IU daily	67	51 (76.1)	31.1 (10.85)	49.71 (27.05)
Sikaroudi et al., 2020 [30]	Iran	Single-center double blinded RCT	88	IBS-D	2	44	25 (56.8)	35.07 (11.73)	17.68 (7.69)	50,000 IU weekly	44	22 (50)	35.61 (8)	17.83 (7.84)
Jalili et al., 2019 [29]	Iran	Multi-center Double blinded RCT	116	N/A	1.5	58	58 (100)	52.24 (12.26)	N/A	50,000 IU weekly	58	58 (100)	40.06 (13.37)	N/A
El Amrousy et al., 2018 [28]	Egypt	Single-center double blinded RCT	112	IBS-C IBS-U IBS-M IBS-D	6	56	29 (52)	16.4 (1.5)	17.2 (1.3)	2000 IU daily	56	33 (59)	16.2 (1.1)	17.5 (1.1)

RCT: randomized controlled trial, SD: standard deviation, N/A: not available; IU: international unit.

**Table 3 nutrients-14-02618-t003:** GRADE evidence profile: vitamin D compared to placebo for IBS.

Certainty Assessment							No. of Patients		Effect		Certainty	Importance
No. of Studies	Study Design	Risk of Bias	Inconsistency	Indirectness	Imprecision	Other Considerations	Vitamin D	Placebo	Relative (95% CI)	Absolute (95% CI)		
IBS-SSS												
6	randomized trials	not serious	very serious ^a^	not serious	very serious ^b^	none	310	306	-	MD 45.86 lower (93.65 lower to 1.93 higher)	⨁◯◯◯ Very low	Critical
IBS-Qol												
4	randomized trials	serious ^c^	very serious ^a^	not serious	serious ^d^	none	226	222	-	MD 6.19 higher (0.35 higher to 12.03 higher)	⨁◯◯◯ Very low	Important
Serum 25(OH)D												
4	randomized trials	not serious	very serious ^a^	serious ^e^	not serious	very strong association	168	164	-	MD 30.03 higher (20.72 higher to 39.34 higher)	⨁⨁⨁◯ Moderate	Important

CI: confidence interval; MD: mean difference; ⨁: positive evidence of certainty. ^a^ I square test is more than 60 percent. ^b^ The 95% confidence interval does not exclude the null hypothesis of 0 mean difference. ^c^ Abbasnezhad et al. [27] is at a high risk of attrition bias and represents 42 percent of the pooled analysis weight. ^d^ The 95% confidence interval (CI) does not exclude the MD of a 0.5 with a very wide CI. ^e^ Despite the large effect and significant CI the total number of participants is less than 400.

**Table 4 nutrients-14-02618-t004:** Sensitivity analysis.

Outcome	No. of participants (Vitamin D/Placebo)	No. of Trials	Quantitative Data Synthesis				Heterogeneity Analysis		
			MD	95% CI	Z Value	*p*-Value	df	*p*-Value	I2 (%)
**IBS-SSS**									
All studies	310/306	6	−45.82	[−93.62, 1.98]	1.88	0.06	5	0.00001	95
Omitting Abbasnezhad et al., 2016 [27]	266/265	5	−47.23	[−118.33, 23.88]	1.3	0.19	4	0.00001	95
Omitting El Amrousy et al., 2018 [28]	254/250	5	−42.28	[−99.7, 15.13]	1.44	0.15	4	0.00001	96
Omitting Jalili et al., 2019 [29]	252/248	5	−59.82	[−111.85, −7.79]	2.25	0.02	4	0.00001	95
Omitting Sikaroudi et al., 2020 [30]	266/262	5	−43.01	[−98.52, 12.49]	1.52	0.13	4	0.00001	96
Omitting Williams et al., 2021 [31]	242/239	5	−57.82	[−110.94, −4.71]	2.13	0.03	4	0.00001	95
Omitting Zeid et al., 2020 [32]	270/266	5	−24.33	[−54.85, 6.19]	1.56	0.12	4	0.0003	81
**IBS-Qol**									
All studies	226/222	4	6.19	[0.35, 12.03]	2.08	0.04	3	0.03	66
Omitting Abbasnezhad et al., 2016 [27]	182/181	3	8.23	[0.15, 16.31]	2	0.05	2	0.13	51
Omitting El Amrousy et al., 2018 [28]	170/166	3	3.26	[2.14, 4.39]	5.67	0.00001	2	0.98	0
Omitting Jalili et al., 2019 [29]	168/164	3	6.41	[−0.37, 13.18]	1.85	0.06	2	0.01	77
Omitting Williams et al., 2021 [31]	158/155	3	7.41	[−0.96, 15.77]	1.73	0.08	2	0.01	77
**Serum 25(OH)D**									
All studies	226/222	4	25.20	[18.41, 31.98]	7.28	0.00001	3	0.00001	93
Omitting Abbasnezhad et al., 2016 [27]	182/181	3	22.58	[15.39, 29.76]	6.16	0.00001	2	0.0002	88
Omitting El Amrousy et al., 2018 [28]	170/166	3	28.16	[14.87, 41.46]	4.15	0.0001	2	0.00001	95
Omitting Williams et al., 2021 [31]	158/155	3	22.54	[15.80, 29.29]	6.55	0.00001	2	0.00001	94
Omitting Jalili et al., 2019 [29]	168/164	3	29.33	[20.29, 38.38]	6.36	0.00001	2	0.00001	94

CI: confidence interval; df: degrees of freedom; MD: mean difference.

## Data Availability

Not applicable.

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
