# Peer review of "The Effect of Vitamin D Supplementation on the Severity of Symptoms and the Quality of Life in Irritable Bowel Syndrome Patients: A Systematic Review and Meta-Analysis of Randomized Controlled Trials"

_nutrients, 2022, doi:10.3390/nu14132618_

Round 1
Reviewer 1 Report
This is a properly conducted SLR, that followed the PROSPERO registered protocol [CRD42022323299] and reported as per PRISMA. Findings are important as the current evidence from the primary study is conflicting. I have provided a few suggestions below:
1. Number of participants mentioned in the abstract and main result section are conflicting.
2. Revisit the below line, calculation not adding up
"a total population of 641 participants including 310 in the vitamin D group and 306 in the placebo"
3. In Table 2. also mention if the trials were single centre or multicenter.
4. Use abbreviation one only, eg: PRISMA in line 140
Author Response
Reviewer 1: Comments and Suggestions for Authors
This is a properly conducted SLR, that followed the PROSPERO registered protocol [CRD42022323299] and reported as per PRISMA. Findings are important as the current evidence from the primary study is conflicting. I have provided a few suggestions below:
We thank Reviewer 1 for their comments.
- Number of participants mentioned in the abstract and main result section are conflicting.
Thank you so much for pointing this out. We revised this concern and edited the total number of participants in the main results accordingly. The correct total number is 616 participants.
- Revisit the below line, calculation not adding up
"a total population of 641 participants including 310 in the vitamin D group and 306 in the placebo"
Thank you so much for pointing this out. We revised this concern and edited the total number of participants in the main results accordingly. The correct total number is 616 participants.
- In Table 2. also mention if the trials were single centre or multicenter.
Thank you for this suggestion. We edited Table 2 accordingly.
- Use abbreviation one only, eg: PRISMA in line 140.
Thank you so much for pointing this out. We edited line 140 accordingly.

Reviewer 2 Report
The article presented by Mohamed Abuelazm and collaborates, entitled “Effect of Vitamin D Supplementation on the Severity of Symptoms and the Quality of Life in Irritable Bowel Syndrome Patients: A Systematic Review and Meta-Analysis of Randomized Controlled Trials”, is a systematic review that aimed to update and summarize the evidence on the real benefits of vitamin D supplementation in the clinical improvement of Irritable Bowel Syndrome. Nutritionally, it is a topic that arouses a lot of interest in patients diagnosed with IBS and it is interesting to know the results obtained by the article, where they give a global and objective view of the possible benefits of vitamin D supplementation. The work is well designed, the search terms and the databases used are correct. Figure 1 is very illustrative and highlights the few articles where the experimental design is correct (adequate controls, etc.), since the systematic review is only carried out with 6 articles of the 24 potential ones. The stated objectives are coherent and are answered throughout the work. The introduction is correct, but I would describe a little the Bowel Severity Scoring System (IBS-SSS) and IBS quality of life (IBS-QoL).
Major revision:
1. It is necessary to include a funnel plot
2. It is necessary to include the forest plot of IBS-Qol excluding the study of El Amrousy (line 34 page 12) because the authors rely on that data to reach their conclusions
Minor revision
1. Line 55: “Worldwide, a 30-50% prevalence has been estimated for vitamin D deficiency [19] 55 and gastrointestinal diseases”. It is not clear from the text if the 30-50% prevalence is only vitamin D deficiency or vitamin D deficiency in patients with gastrointestinal pathology
2. Line 173: UK 31]
3. Line 41: e3ach
4. Line 49: Egypt)
Author Response
Reviewer 2: Comments and Suggestions for Authors
The article presented by Mohamed Abuelazm and collaborates, entitled “Effect of Vitamin D Supplementation on the Severity of Symptoms and the Quality of Life in Irritable Bowel Syndrome Patients: A Systematic Review and Meta-Analysis of Randomized Controlled Trials”, is a systematic review that aimed to update and summarize the evidence on the real benefits of vitamin D supplementation in the clinical improvement of Irritable Bowel Syndrome. Nutritionally, it is a topic that arouses a lot of interest in patients diagnosed with IBS and it is interesting to know the results obtained by the article, where they give a global and objective view of the possible benefits of vitamin D supplementation. The work is well designed, the search terms and the databases used are correct. Figure 1 is very illustrative and highlights the few articles where the experimental design is correct (adequate controls, etc.), since the systematic review is only carried out with 6 articles of the 24 potential ones. The stated objectives are coherent and are answered throughout the work. The introduction is correct, but I would describe a little the Bowel Severity Scoring System (IBS-SSS) and IBS quality of life (IBS-QoL).
We thank Reviewer 1 for their comments.
We appreciate you discussing this point with us. We have described the Bowel Severity Scoring System (IBS-SSS) and IBS quality of life (IBS-QoL) in the methods section under eligibility criteria sub-section 2.3 from line 92 to 101 as follows:
The IBS-SSS is a questionnaire validated for application for IBS patients to determine the burden of the disease; during 10 days, IBS-SSS assesses the following (abdominal pain severity - abdominal pain frequency – abdominal distention or tightness severity – bowel habits dissatisfaction – interference with life in general) each item is scored on a scale from 0 to 100 and with a range of (0 – 500). Mild, moderate, and severe cases are indicated by scores of 75–175, 175–300, and >300, respectively, and a score re-duction of 50 or more is considered clinically significant [35]. The (IBS-Qol) questionnaire included dysphoria, interference with activity, body image, health worry, food avoidance, social reaction, sexual, and other issues among eight subscales and 34 items with a score range from 0 to 100; the higher the score the better Qol [36].
References:
- Francis, C.Y.; Morris, J.; Whorwell, P.J. The Irritable Bowel Severity Scoring System: A Simple Method of Monitoring Irritable Bowel Syndrome and Its Progress. Aliment. Pharmacol. Ther. 1997, 11, 395–402. https://doi.org/10.1046/j.1365-2036.1997.142318000.x
- Drossman, D.A.; Patrick, D.L.; Whitehead, W.E.; Toner, B.B.; Diamant, N.E.; Hu, Y.; Jia, H.; Bangdiwala, S.I. Further Validation of the IBS-QOL: A Disease-Specific Quality-of-Life Questionnaire. Am. J. Gastroenterol. 2000, 95, 999–1007.
Major revision:
- It is necessary to include a funnel plot
Thank you for your suggestion. We followed Egger et al.’s recommendations by not performing a funnel plot because our review included less than 10 studies.
We included an explanation under 4.2. Limitations as follows:
Because we had fewer than ten studies, we did not perform a funnel plot [44].
- Egger, M.; Smith, G.D.; Schneider, M.; Minder, C. Bias in Meta-Analysis Detected by a Simple, Graphical Test. BMJ 1997, 315, 629–634. https://www.bmj.com/content/315/7109/629
- It is necessary to include the forest plot of IBS-Qol excluding the study of El Amrousy (line 34 page 12) because the authors rely on that data to reach their conclusions
Thank you for your suggestion. We edited our results section accordingly by adding Figure 5 Forest plot of IBS-Qol after exclusion El-Amrousy et al. [27].
Minor revision
- Line 55: “Worldwide, a 30-50% prevalence has been estimated for vitamin D deficiency [19]55 and gastrointestinal diseases”. It is not clear from the text if the 30-50% prevalence is only vitamin D deficiency or vitamin D deficiency in patients with gastrointestinal pathology
Thank you for criticizing this point. We edited it accordingly as follows: Worldwide, a 30-50% prevalence has been estimated for vitamin D deficiency across different demographic groups.
- Line 173: UK 31]
Thank you for pointing this out. We addressed this accordingly.
- Line 41:e3ach
Thank you for pointing this out. We addressed this accordingly.
- Line 49: Egypt)
Thank you for pointing this out. We addressed this accordingly.

Round 2
Reviewer 2 Report
The authors have answered my questions and made the appropriate corrections